# A 10-Year Retrospective Analysis of Medication Errors among Adult Patients: Characteristics and Outcomes

**DOI:** 10.3390/pharmacy11050138

**Published:** 2023-09-01

**Authors:** Phantakan Tansuwannarat, Piraya Vichiensanth, Ornlatcha Sivarak, Achara Tongpoo, Puangpak Promrungsri, Charuwan Sriapha, Winai Wananukul, Satariya Trakulsrichai

**Affiliations:** 1Chakri Naruebodindra Medical Institute, Faculty of Medicine Ramathibodi Hospital, Mahidol University, Bangkok 10400, Thailand; phantakan.tans@gmail.com; 2Ramathibodi Poison Center, Faculty of Medicine Ramathibodi Hospital, Mahidol University, Bangkok 10400, Thailand; achara.ton@mahidol.ac.th (A.T.); puangpak.prom@gmail.com (P.P.); charuwan.sri@mahidol.ac.th (C.S.); winai.wan@mahidol.edu (W.W.); 3Department of Emergency Medicine, Faculty of Medicine Ramathibodi Hospital, Mahidol University, Bangkok 10400, Thailand; mooky.pv@gmail.com; 4International College, Mahidol University, Nakhon Pathom 73170, Thailand; ornlatcha.siv@mahidol.ac.th; 5Department of Medicine, Faculty of Medicine Ramathibodi Hospital, Mahidol University, Bangkok 10400, Thailand

**Keywords:** medical error, deaths, hospitals, adults, iatrogenic, local anesthetic

## Abstract

Medication errors (MEs) are a global health problem. We conducted this study to clarify the clinical characteristics, outcomes, and factors associated with MEs that caused harm to adult patients (>15 years of age) who were managed in hospitals or healthcare facilities. We performed a 10-year retrospective study (2011–2020) by analyzing data from the Ramathibodi Poison Center (RPC) database (RPC Toxic Exposure Surveillance System). There were a total of 112 patients included in this study. Most were women (59.8%) and had underlying diseases (53.6%). The mean patient age was 50.5 years. Most MEs occurred during the afternoon shift (51.8%) and in the outpatient department (65.2%). The most common type of ME was a dose error (40.2%). Local anesthetic was the most common class of ME-related drug. Five patients died due to MEs. We analyzed the factors associated with MEs that caused patient harm, including death (categories E–I). The presence of underlying diseases was the single factor that was statistically significantly different between groups. Clinical characteristics showed no significant difference between patients aged 15–65 years and those aged >65 years. In conclusion, our findings emphasized that MEs can cause harm and even death in some adult patients. Local anesthetics were the most commonly involved in MEs. Having an underlying disease might contribute to severe consequences from MEs. Preventive measures and safety systems must be highlighted and applied to prevent or minimize the occurrence of MEs.

## 1. Introduction

Medications are a key element in treating many diseases and health problems. However, medications can cause harm if errors occur. A medication error (ME) is any preventable event or failure in the treatment process that introduces a patient to inappropriate medication use or patient harm during medication treatment that is under the control of healthcare providers, patients, or consumers [1,2,3]. MEs can occur at each point in the medication management process, owing to prescribing errors, dispensing errors, administration errors, or transcribing errors [4,5].

MEs are reported worldwide, including in Southeast Asia [6,7,8]. Several risk factors for the occurrence of MEs have been identified [6,9]. One patient-related risk factor is older age [6,9,10,11], with studies reporting that older patients are at increased risk of MEs. Patients aged more than 65 years experience approximately three-times more MEs than those younger than 65 years of age [11]. Older adults are a group at high risk of MEs, and some patient-related factors, such as polypharmacy and comorbidities, might contribute to this risk [11]. The consequences of MEs involve morbidity and mortality [6,7,8,12,13]. Several patients develop serious clinical effects and sequelae from MEs. Apart from the clinical consequences of iatrogenic MEs occurring within the hospital system, MEs can affect patients, their relatives, and medical personnel by increasing hospital expense, prolonging the length of stay in a hospital, and contributing to psychological effects [12,14]. Therefore, healthcare facilities and hospitals should consider implementing preventive measures and applying medication safety systems to prevent patient harm that is caused by MEs.

With different socioeconomic backgrounds or healthcare systems among various countries or continents, MEs may differ in terms of clinical incidence or characteristics. MEs that occur in Southeast Asian countries were described in a systematic review [7], and several studies have reported MEs in Thai patients [9,15,16,17].

One study showed that data on MEs from poison information centers could help to characterize MEs that occur in both community settings and healthcare facilities, which could serve to heighten pharmacovigilance [14,18]. In the abovementioned systematic review reporting MEs in Southeast Asian countries, none of the included studies used poison center data, and the clinical outcomes of MEs were not aggressively examined in most studies [7].

In Thailand, several cases of MEs occurring in both community and healthcare settings have referred and consulted voluntarily to the Ramathibodi Poison Center (RPC) in Bangkok. The RPC is a poison control center based in a tertiary teaching hospital, the Faculty of Medicine Ramathibodi Hospital, Mahidol University in Thailand. The consultations to the RPC are from every region in our country. The RPC might be able to provide data on the clinical characteristics and outcomes of MEs occurring across the healthcare system in Thailand. The present study was performed to describe and clarify the clinical characteristics, outcomes, and factors associated with MEs occurring in hospitals that have caused patient harm, including fatality (categories E–I based on the ME taxonomy developed by the National Coordinating Council for Medication Error Reporting and Prevention [19]). We also compared clinical characteristics between patients aged 15–65 years and patients aged >65 years.

## 2. Materials and Methods

### 2.1. Study Design

We performed a 10-year retrospective study (January 2011–December 2020) by reviewing and analyzing data from the RPC database (RPC Toxic Exposure Surveillance System). The primary outcomes were the clinical characteristics, management, and outcomes of adult patients with MEs during their treatments in the hospitals. The secondary outcomes were factors associated with MEs that caused harm, including death (categories E–I), in adult patients and the differences in clinical characteristics between patients aged 15–65 years and those aged >65 years.

The present study was approved by the Institutional Ethics Committee Board of Ramathibodi Hospital, Faculty of Medicine, Mahidol University (COA. MURA2021/229). The requirement for patient informed consent was waived by our hospital’s ethics committee according to the retrospective design of this study and the anonymized reporting of the confidential data derived from the poison center database.

### 2.2. Study Setting and Population

The setting was the RPC located in Bangkok, Thailand. The RPC receives approximately 20,000–30,000 consultations per year, mostly from medical personnel, and provides information on clinical assessment, diagnosis, and management, including the monitoring and disposition of cases of poisoning. Immediate access is available 24 h a day for both medical personnel and the general public. All inquiries are received and answered by specialists in poison information (SPIs). These specialists are pharmacists or nurses who must complete training and pass all compulsory exams to be certified as being competent in the knowledge, skills, and experience needed to provide poison-related information to medical personnel and the general population. For uncertain or complicated cases, consultations with medical toxicologists who serve as consultants to provide additional medical backup to SPIs are required and available. Follow-up telephone calls are performed to collect patient data and information on patients’ progress to provide additional treatment recommendations and to determine medical outcomes. All patients’ data are recorded in the RPC Toxic Surveillance System database. A team of senior SPIs and medical toxicologists verifies and conducts a final review of consultations, particularly the final diagnosis and outcome.

The inclusion criteria in this study were patients aged >15 years who experienced the occurrence of a therapeutic error that was defined in the RPC Toxic Exposure Surveillance System, together with a ME caused by healthcare personnel and took place in a hospital setting from January 2011 to December 2020. The term “therapeutic error” indicated patients who had “an exposure (or incident) resulting from the incorrect use of medication, whether the agent was administered by medical personnel or by a lay person.” [20]. Patients with incomplete data for outcomes were excluded from the study.

The following data were collected for all patients: demographic data, underlying diseases, clinical characteristics, types of ME, types of drugs associated with ME, management, and outcomes. The clinical characteristics, diagnosis, and outcomes of all patients recruited in this study were reviewed and verified by clinical toxicologists.

### 2.3. Definitions

The definition of MEs in this study was errors during drug prescribing, transcribing, administering, dispensing, and monitoring that occurred during hospitalization in either outpatient or inpatient settings. Administration errors (the difference between what the patient received and what the prescriber intended in the original order), route errors (an incorrect route of administration), and dose errors (differences in dose, strength, quantity, or frequency from the standard ones) were defined according to a previous study definition [12]. Administration errors in this study included a patient error, which occurred when medication was given to the wrong patient.

MEs are classified into categories A to I based on the severity of the outcome [19]. Categories C and D are errors that affect patients but do not cause harm and those that affect patients and require monitoring and/or intervention to prevent harm, respectively. Categories E–I are errors that affect patients and cause temporary to permanent patient harm, including death [12,19].

Hypotension was determined as systolic blood pressure less than 90 mmHg [21]. High blood pressure was defined as 140/90 mmHg or higher [22]. Bradycardia and tachycardia were stated as heart rate less than 60 and more than 100 beats per minute, respectively [23]. Fever was identified as body temperature higher than 37.7 °C [24]. Tachypnea and bradypnea referred to faster or slower rates of breathing than normal for an average adult (12–20 breaths/minute) [25]. Acute kidney injury was determined by the Kidney Disease: Improving Global Outcomes clinical practice guidelines [26]. We assumed that patients with no history of kidney disease previously had normal kidney function prior to ME. Hyponatremia and hypernatremia were determined as serum sodium less than 135 and more than 145 mEq/L, respectively. Hypokalemia and hyperkalemia were determined as serum potassium less than 3.5 and more than 5.0 mEq/L, respectively [27].

### 2.4. Statistical Analysis

In this study, Stata version 17 (StataCorp, College Station, TX, USA) was used to analyze the data. Continuous data were described as mean and standard deviation if the data were normally distributed; otherwise, the data were described as median, with minimum and maximum. Categorical data are shown as frequency and percentage. Comparisons between groups are made using the Student *t*-test if two independent continuous datasets are normally distributed; otherwise, we use the Mann–Whitney U-test. Differences in categorical variables are analyzed by using chi-squared analysis and the Fisher’s exact test.

## 3. Results

A total of 1033 patients were recorded as experiencing therapeutic errors. Eight hundred and forty-three patients experiencing MEs who did not meet the criteria were excluded. Therefore, iatrogenic MEs that occurred in a hospital setting were identified in 18.4% of the total patients. The remaining errors occurred in a community setting. We excluded patients aged <15 years, leaving a total of 112 patients who met the inclusion criteria and were included in the analysis.

Patient demographic data are presented in Table 1. Most patients (59.8%) were women. The median patient age was 50.5 years old (range, 15–99 years). There were more adult patients aged <65 years than older patients in our study. MEs were most commonly reported in the northeast region of Thailand (Figure 1). Most MEs occurred during the afternoon shift (58; 51.8%) and were most frequently in an outpatient department (65.2%). Errors in the medication process, such as prescribing or dispensing, were not recorded in our database; therefore, these data were not included in this study.

Table 2 describes the type of ME and class of drugs for the reported MEs. The most common type of ME was incorrect dose (45 patients, 40.2%). The most common drugs with a wrong dose error were lidocaine (six patients) and heparin (four patients). The most common drugs involved in a wrong route error were ipratropium bromide/fenoterol (from nebulized to intravenous route; eight patients), and diclofenac (from intramuscular to intravenous route; seven patients). The most common wrong drug error involved the administration of lidocaine instead of glucose (eight patients), methotrexate instead of multivitamin (two patients), and lidocaine instead of amoxicillin/clavulanate (one patient).

We analyzed classes of drugs according to the medications received by patients. Local anesthesia (13.4%), antipsychotic drugs (8.9%), cardiology drugs (8.9%), and bronchodilators (8.0%) were the most common drug types involved in MEs reported in our study.

The most common local anesthetic drug related to MEs was lidocaine (nine patients with a wrong drug error and six patients with a wrong dose error). Four patients with MEs involving lidocaine developed cardiac arrest, and three of them died. One 74-year-old man who survived received intravenous lidocaine instead of glucose. The most common antipsychotic drugs involved in MEs were fluphenazine (four patients with a wrong dose error), clozapine (two patients with a wrong dose error and one patient with a wrong drug error), haloperidol (two patients with wrong dose and route errors), and chlorpromazine (one patient with a wrong drug error). The most common cardiology drugs in MEs were digoxin (two patients with a wrong drug error) and the other drugs included sodium nitroprusside, amiodarone, adrenaline, diltiazem, verapamil, and norepinephrine (one patient each). ME-related bronchodilators included fenoterol/ipratropium bromide (eight patients with a wrong route error) and salbutamol (one patient with a wrong route error). Among antineoplastic drugs, methotrexate was the most commonly related to MEs (three patients with a wrong dose error and two patients with a wrong drug error).

Figure 2 demonstrates the patient’s category of error. ME category F occurred most frequently (50.9%). Of 112 patients, 5 patients (4.5%) were identified as ME category H, and 5 patients (4.5%) were identified as ME category I.

Table 3 describes the clinical features at the first presentation of MEs. Most patients had normal vital signs, Glasgow Coma Scale, and oxygen saturation in pulse oximetry. Tachycardia after MEs occurred in approximately 30% of the total patients. The involvement of the neurological system was most common after MEs in our study, with drowsiness as the most common effect (15 patients).

Table 4 presents laboratory results and investigations after MEs; normal electrocardiogram (EKG) was the most common finding. Two patients developed cardiac arrest, with the EKG showing asystole; three patients developed ventricular tachycardia (VT).

Intravenous fluids and administration of activated charcoal were the two most commonly used treatment modalities. Thirteen patients (11.6%) required endotracheal intubation, eleven (9.8%) needed inotropic drugs/vasopressors, and six patients (5.4%) required cardiopulmonary resuscitation (CPR) (Table 5).

Five patients with MEs died (Table 6); thus, the mortality rate in our study was 4.5%. All five dead patients were female. Their age ranged from 38 to 78 years old. Three had underlying diseases. Their MEs occurred in the afternoon shift (three patients), morning shift (one patient), and night shift (one patient). Three patients received wrong doses of lidocaine, including injection (400 mg instead of 100 mg) via the pleural cavity, intravenous (IV) injection (1000 mg instead of 50 mg), and IV injection (400 mg instead of 20 mg) to treat VT. All three patients developed generalized tonic-clonic seizure, status epilepticus, and cardiac arrest. Two patients were treated with 20% fat emulsion as the antidote and standard resuscitation measures. One dead patient received the wrong drug (IV injection of 2 mg digoxin (0.5 mg, 4 times)) instead of the required total dose of 2.4 mg atropine. The other ME that caused death occurred from a wrong route error by giving benzathine benzylpenicillin intravenously instead intramuscularly (three doses every 4 h). Four patients died within a few hours (range, 1.5–3.5 h), while one died 36 h after MEs occurred.

To analyze the factors associated with MEs leading to patient harm (categories E–I), we compared clinical characteristics between patients with MEs categories C–D and those with categories E–I (Table 7). Having an underlying disease was significantly different between both groups of patients (Table 7). We also compared clinical characteristics between patients aged <65 years and those aged >65 years, as shown in Table 8. No significant difference was found between the two groups.

## 4. Discussion

MEs have been reported in many countries worldwide [6,7,8,10,28], in both pediatric and adult patients [7,10,12,28,29,30]. However, there are limited data from poison centers describing MEs in Asia, especially Southeast Asia [7,12,31,32,33,34,35]. In this study, we describe the patient characteristics and outcomes of MEs, including MEs with severe and fatal outcomes, among Thai adult patients over a 10-year period using data reported to the RPC. Our data can help characterize MEs occurring across healthcare facilities, particularly with respect to common MEs that occur in Thailand. Our findings represent the epidemiology, clinical effects, and consequences of MEs in all regions of Thailand. MEs are reported in both hospital settings [5,7,9,12] and domestic settings, as well as in outpatient healthcare facilities [10,11,36]. In this study, we focused on MEs occurring among adult patients only in healthcare facilities and that were caused by medical personnel. MEs that occur in hospitals or healthcare facilities can harm patients, and our data can help guide these facilities in improving safety systems. All MEs caused by healthcare workers are usually preventable. The demographic data of our patients demonstrated that MEs occur throughout Thailand and among women more than men. Our findings are consistent with those of MEs reported to poison control centers in the United States, showing that MEs commonly occur in adults over 20 years of age and mostly in women [10]. In our study, adult patients aged 15–65 years experienced MEs more frequently than older patients. We found that MEs most commonly occurred in the northeast region of Thailand, followed by central Thailand. The work shift during which MEs occurred most commonly was the afternoon shift, followed by the morning shift. These results differed from some studies of MEs in Iran [37,38], where the highest average number of MEs occurred during the night shift. Differences among the study findings might be due to different backgrounds in terms of healthcare systems or facilities in different countries, including the number of clinical staff, volume of work, and type of patient services provided. However, the finding that MEs occurred mostly during the afternoon shift among adult patients was consistent with our previous study among pediatric patients [12]; therefore, in Thailand, greater attention to safety measures is required, especially during the afternoon shift. Further study is needed to clarify this finding.

We found that MEs in our study occurred most frequently in the ED and outpatient departments, which was different from our finding among pediatric patients, that MEs mostly occurred in inpatient departments [12]. This difference might be owing to differences in the study populations or treatment settings for each patient population. On the basis of our present study findings, greater focus is needed on healthcare safety education or preventive measures in the ED and in outpatient departments. The most common classes of drugs reported in our study were local anesthesia, antipsychotic drugs, and cardiology drugs. Our findings are consistent with the results of other previous studies, showing that these are common drugs associated with MEs [9,13]. Interestingly, local anesthesia was the most common drug involved in MEs, and all MEs related to local anesthesia involved lidocaine. MEs in nine patients were due to the administration of the wrong drug, with patients who required glucose (eight patients) and amoxicillin/clavulanate (one patient) injections incorrectly given lidocaine injections instead. This ME might be partly explained by confusion caused by drug containers that look similar (especially look-alikes to the vial of 50% hypertonic glucose solution), which can easily be confused. This error poses a systematic threat to patient safety and should be highlighted as an area for urgent improvement [39]. Potential solutions for look-alike vial errors might include evaluation of drug containers prior to drug procurement; avoiding multiple containers for the same medication; labeling high-risk medications; using prefilled syringes whenever possible and barcode scanning; providing pictures or information on look-alike vials; and promoting a culture of safety [39].

Six patients experienced MEs due to receiving an incorrect dose of lidocaine. Three of these patients developed status epilepticus and eventually died from severe lidocaine toxicity; two of these patients received lipid emulsion therapy as an antidote. Severe lidocaine poisoning can cause systemic local anesthetic toxicity and death. The main clinical effects include central nervous system and cardiovascular toxicities [40]. This highlights the importance of lidocaine toxicity from MEs in Thailand and should encourage the promotion of safety measures for this drug. Cardiology drugs and antipsychotic drugs were the second most common drugs related to MEs, followed by bronchodilators. According to our previous study of MEs in pediatric patients, bronchodilators and cardiology drugs were the third and fourth most common drugs reported in relation to MEs. These findings might be owing to the types of drugs most commonly used for the treatment of patients in each age group.

MEs were frequently caused by errors involving a wrong dose and/or the wrong drug. Our finding regarding dose error was consistent with that of other studies [8,12,13,35], especially regarding deaths related to iatrogenic errors, mostly involving an incorrect dose of a drug.

Most patients in this study had category D to I MEs, and category F was commonly found in approximately half of our patients. Therefore, most patients had prolonged hospitalization owing to MEs. MEs increase the cost of hospital treatment and might contribute to hospital bed shortages.

We found that the factor associated with ME-induced patient harm was having an underlying disease. However, we found no significant difference in clinical characteristics between patients aged <65 years and those aged >65 years.

Our findings highlighted that MEs occurring in hospitals can contribute to severe consequences and fatal outcomes in adult patients. Most MEs reported in our study occurred in female patients, during the afternoon and morning shifts, and were owing to the administration of a wrong dose or the wrong drug, especially involving lidocaine. It is important to accentuate and increase awareness among medical personnel regarding MEs and to implement safety measures and systems to give the correct medication in the correct dosage form, at the correct dose, via the correct route, to the correct patient, and at the correct time. Safety systems and measures can include educational strategies to improve knowledge and the use of new organizational technologies such as patient prescription using a computerized system [2,41].

This study had the following limitations. First, the true incidence of MEs might be underestimated because MEs are only voluntarily consulted and reported to the RPC. Second, ME categories A and B were not included in our study because these categories do not affect patients, so physicians do not report such cases to the RPC. Third, because we could not collect data on the total prescriptions in each hospital, we could not analyze the overall rate of MEs per total prescriptions. Fourth, the study was performed retrospectively; therefore, this study design may have resulted in missing or incomplete data or reporting bias. Finally, we could not retrieve data on MEs that occurred during each processing stage, such as dispensing or administration, to guide and help improve the safety system at each processing stage. SPIs might not actively query healthcare professionals, who may feel uncomfortable describing their error. Further investigation is required to address these limitations.

## 5. Conclusions

Our findings from the data of the RPC indicated that most MEs among adult patients occurred during the afternoon shift owing to a wrong medication dose. Our data emphasized that MEs caused severe outcomes and even death in some patients. Greater attention is urgently needed to implement safety measures to avoid MEs, such as the administration of local anesthetic via the incorrect route. Addressing common causes of MEs, such as errors owing to look-alike vials, which can result in administration of the wrong drug, included preventive measures in the safety system that should be emphasized and implemented to prevent or minimize the occurrence of MEs. ME reduction strategies should be accentuated in hospitals to improve the quality of medical care and patient safety.

## Figures and Tables

**Figure 1 pharmacy-11-00138-f001:**
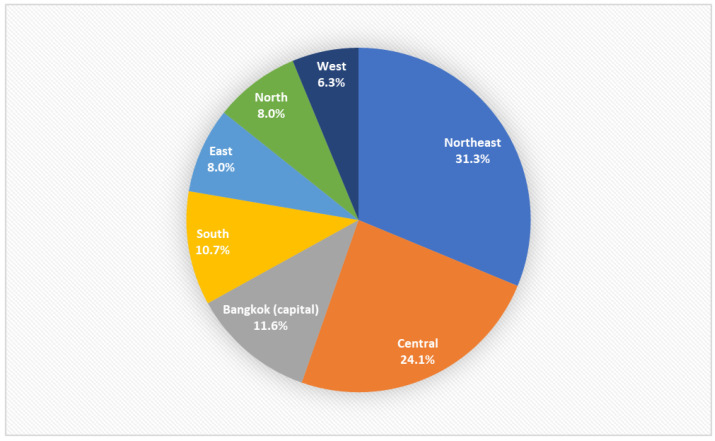
Regions of the consultations.

**Figure 2 pharmacy-11-00138-f002:**
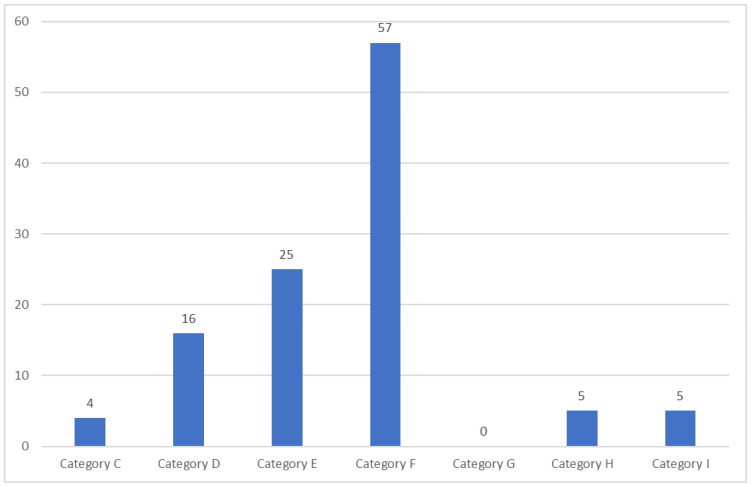
The category of error of all patients.

**Table 1 pharmacy-11-00138-t001:** Demographic data of patients.

Variables	Number (%)
**Sex**	
Male	45 (40.2)
Female	67 (59.8)
**Age (year),** mean ± SD	50.5 (20.7)
**Age group**	
15–65 years	81 (72.3)
>65 years	31 (27.7)
**Type of hospital**	
Government	100 (89.3)
Private	12 (10.7)
**Work shift**	
Afternoon	58 (51.8)
Morning	50 (44.6)
Night	4 (3.6)
**Underlying diseases**	
Yes	60 (53.6)
No	52 (46.4)
**Location of patient management**	
Emergency room/Outpatient department	73 (65.2)
Inpatient department (wards and intensive care units)	39 (34.8)

SD, standard deviation.

**Table 2 pharmacy-11-00138-t002:** Types of medication error and class of drugs, involved in medication errors.

Variables	Number (%)
**Type of medication error**	
Wrong dose	45 (40.2)
Wrong drug	34 (30.4)
Wrong route	26 (23.2)
Wrong administration	7 (6.3)
**Class of drugs**	
Local anesthetics	15 (13.4)
Cardiology drugs	10 (8.9)
Antipsychotic drugs	10 (8.9)
Bronchodilators	9 (8.0)
Antineoplastic drugs	8 (7.1)
Antiepileptic drugs	7 (6.3)
Analgesics: non-steroidal anti-inflammatory drugs	7 (6.3)
Antibiotics	5 (4.5)
Antivirals	5 (4.5)
Anticoagulants	4 (3.6)
Opioids	4 (3.6)
Antidiabetic drugs	3 (2.7)
Antidotes	2 (1.8)
Antihistamines	2 (1.8)
Other (each for 1 case)	21 (18.8)

**Table 3 pharmacy-11-00138-t003:** Clinical features when medication error occurred.

Variables	Number (%)
**Pulse rate**	
Normal	74 (66.0)
Tachycardia	33 (29.5)
Bradycardia	5 (4.5)
**Blood pressure**	
Normal	100 (89.3)
Shock	12 (10.7)
**Respiratory rate**	
Normal	109 (97.3)
Bradypnea	3 (2.7)
**Body temperature**	
Normothermia	101 (90.2)
Hyperthermia	11 (9.8)
**Glasgow Coma Scale**	
15	97 (86.6)
<15	15 (13.4)
**Oxygen saturation from pulse oximeter**	
≥95%	109 (97.3)
<95%	3 (2.7)
**Systems involvement** (some patients had >1 system and/or >1 symptom)	
Neurological system ^a^	38 (33.9)
Respiratory system ^b^	13 (11.6)
Gastrointestinal system ^c^	11 (9.8)
Dermatological system ^d^	3 (2.7)
Cardiovascular system ^e^	1 (0.9)

^a^ Drowsiness (15 patients), unconsciousness (12 patients), seizure (8 patients), tremor (1 patient), dizziness (3 patients), headache (1 patient), oral numbness (1 patient), paresthesia (1 patient), sleep disturbance (1 patient). ^b^ Dyspnea (11 patients), tachypnea (5 patients), bradypnea (3 patients). ^c^ Nausea/vomiting (10 patients), abdominal pain (3 patients), diarrhea (1 patient). ^d^ Phlebitis (2 patients), local inflammation (1 patient). ^e^ Palpitation (1 patient).

**Table 4 pharmacy-11-00138-t004:** Laboratory results of patients in whom medication errors occurred.

Variables	Number (%)
**Serum sodium**	
Hyponatremia	5 (4.5)
Normal	105 (93.8)
Hypernatremia	2 (1.8)
**Serum potassium**	
Hypokalemia	15 (13.4)
Normal	95 (84.8)
Hyperkalemia	2 (1.8)
**Renal function**	
Abnormal ^a^	5 (4.5)
Normal	107 (95.5)
**Chest X-ray**	
Abnormal ^b^	2 (1.8)
Normal	110 (98.2)
**Electrocardiogram**	
Abnormal ^c^	20 (17.9)
Normal	92 (82.1)

^a^ Acute kidney injury. ^b^ Pulmonary edema (2 patients). ^c^ 1st degree AV block (1 patient), atrial fibrillation (3 patients), asystole (2 patients), bradycardia (4 patients), QT prolongation (1 patient), sinus tachycardia (3 patients), ST depression at lead 1 inverted T at V3-V6 (1 patient), supraventricular tachycardia (2 patients), ventricular tachycardia (3 patients).

**Table 5 pharmacy-11-00138-t005:** Management of medication errors.

Variables	Number (%)
Nasogastric lavage	3 (2.7)
Activated charcoal administration	36 (32.1)
Intravenous fluids	36 (32.1)
Oxygen administration	22 (19.6)
Endotracheal intubation	13 (11.6)
Inotrope administration	11 (9.8)
Hemodialysis/peritoneal dialysis	2 (1.8)
Cardiopulmonary resuscitation	6 (5.4)

Some patients received > 1 treatment modality.

**Table 6 pharmacy-11-00138-t006:** Clinical characteristics and laboratory results of patients who died (category I).

Characteristics	Patient 1	Patient 2	Patient 3	Patient 4	Patient 5
**Patient characteristics**					
Sex/age (years)	F/38	F/44	F/38	F/63	F/78
Underlying diseases	Yes	No	Yes	No	Yes
Underlying disease, detail	Breast cancer	-	DM	-	PAF, IHD, DM, CKD, SSS
Location	Ward	Ward	Ward	Ward	ER
Work shift	Afternoon	Afternoon	Afternoon	Morning	Night
Diagnosis	Pleural effusion	TCA toxicity	Ventricular tachycardia	CHF	Cysticercosis
Class of drugs	Local anesthetic drug	Local anesthetic drug	Local anesthetic drug	Anti-arrhythmia drug	Antibiotic
Drug name	Lidocaine	Lidocaine	Lidocaine	Digoxin	Benzathine benzyl penicillin
Type of medication error	Wrong dose	Wrong dose	Wrong dose	Wrong drug	Wrong route
**Clinical manifestations at presentation**					
**GI symptoms** (yes/no)					
Nausea/vomiting, abdominal pain, diarrhea	No, No, No	No, No, No	No, No, No	No, No, No	No, No, No
**Neurologic symptoms** (yes/no)					
Altered consciousness, seizure	Yes, Yes	Yes, Yes	Yes, Yes	No, No	Yes, Yes
**Respiratory symptoms** (yes/no)					
Dyspnea, aspiration	No, No	No, No	Yes, No	No, No	No, No
**Cardiovascular symptoms** (yes/no)					
Arrhythmia, hypotension	Yes, Yes	Yes, Yes	Yes, Yes	Yes, Yes	Yes, Yes
**Time to cardiac arrest** **after ME occurred** (minutes)	10	1	1	45	180
**Treatment** (yes/no)					
Oxygen therapy	Yes	Yes	Yes	Yes	Yes
Endotracheal intubation	Yes	Yes	Yes	Yes	Yes
Intravenous fluid	Yes	Yes	Yes	Yes	Yes
Extracorporeal treatment	No	No	Yes	No	No
Cardiopulmonary resuscitation	Yes	Yes	Yes	Yes	Yes
Antidote	Yes	No	Yes	No	No
Time to antidote after ME (hours)	1.5	-	1	-	-
**Complications during hospitalization** (yes/no)	No	No	No	No	No
Complications, detail	-	-	Septic shock, hyperkalemia	-	-
**Time to death after ME occurred** (hours)	2	3.5	36	1.5	3

GI, Gastrointestinal; ME, Medication error; F, Female; TCA, Tricyclic antidepressants; CHF, Congestive heart failure; PAF, Paroxysmal atrial fibrillation; IHD, Ischemic heart disease; DM, Diabetes mellitus; CKD, Chronic kidney disease; SSS, Sick sinus syndrome; ER, Emergency room.

**Table 7 pharmacy-11-00138-t007:** Factors associated with MEs causing harm to patients (categories E–I).

Variables	Category of Error		*p*-Value
	C–D Number (%)	E–I Number (%)	
**Sex**			0.627
Male	9 (45.0)	36 (39.1)	
Female	11 (55.0)	56 (60.9)	
**Age (year),** mean ± SD	44.8 ± 17.9	51.7 ± 21.1	0.1753
**Age group**			0.162
15–65	17 (85.0)	64 (69.6)	
>65	3 (15.0)	28 (30.4)	
**Underlying diseases**			0.020
Yes	6 (30.0)	54 (58.7)	
No	14 (70.0)	38 (41.3)	
**Work shift**			1.000
Morning	9 (45.0)	41 (44.6)	
Afternoon	10 (50.0)	48 (52.2)	
Night	1 (5.0)	3 (3.2)	
**Type of error**			
Wrong dose	11 (55.0)	34 (37.0)	0.136
Wrong route	3 (15.0)	23 (25.0)	0.399
Wrong drug	5 (25.0)	29 (31.5)	0.565
Wrong administration	1 (5.0)	6 (6.5)	1.000

SD, standard deviation.

**Table 8 pharmacy-11-00138-t008:** Comparison of clinical characteristics between patients ≤65 and >65 years old.

Variables	Category of Error		*p*-Value
	Age ≤ 65 y Number (%)	Age > 65 y Number (%)	
**Sex**			0.815
Male	32 (39.5)	13 (41.9)	
Female	49 (60.5)	18 (58.1)	
**Underlying diseases**			0.311
Yes	41 (50.6)	19 (61.3)	
No	40 (49.4)	12 (38.7)	
**Work shift**			0.077
Morning	35 (43.2)	15 (48.4)	
Afternoon	45 (55.6)	13 (41.9)	
Night	1 (1.2)	3 (9.7)	
**Type of error**			
Wrong dose	31 (38.3)	14 (45.2)	0.506
Wrong route	20 (24.7)	6 (19.3)	0.550
Wrong drug	24 (29.6)	10 (32.3)	0.787
Wrong administration	6 (7.4)	1 (3.2)	0.671

## Data Availability

The data are not available for public access because of patient privacy concerns but are available from the corresponding author upon reasonable request.

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
