# Peer review of "A 10-Year Retrospective Analysis of Medication Errors among Adult Patients: Characteristics and Outcomes"

_pharmacy, 2023, doi:10.3390/pharmacy11050138_

Round 1
Reviewer 1 Report
Iatrogenic illness could result in significant mobility and mortality. Likewise, medication errors (could occur during the process of prescription, dispensing, administration and compliance) is one of the major issue related to the iatrogenic hospitalization and illness. If the total case number collected could be more, it may show us with some different result. However, your dedicated work is appreciated.
Reviewer 2 Report
Dear Authors,
The manuscript, entitled 'A 10-year retrospective analysis of medication errors among adult patients: characteristics and outcomes', presents moderate quality data with numerous limitations on interpretation. Still, I believe it can be published after the introduction of figures and graphs to better illustrate the results (where possible).
Kind reagards.
Reviewer 3 Report
This manuscript, A 10-year retrospective analysis of medication errors among adult patients: characteristics and outcomes, investigated medication errors (MEs) that took place in a hospital setting over the past ten years. While the study is of potential interest, further reworking is needed to move beyond the conclusion that MEs can cause harm and even death, which is already very well known.
I have outlined some specific suggestions below:
Title
Consider amending the title to be clear that only hospital errors were included in this analysis
Abstract
· A bit more about the methods is needed. How was data obtained? From the results presented in the abstract it seems that only MEs caused by health professionals were included?
· 'for the results, there were 112 patients’ – reword
· ‘Six patients died’ – perhaps remove from abstract, or expand – died due to ME?
· Conclusion – consider rewording and better explaining what your study added (it is already well known that MEs can cause harm and even death)
Key words
· Consider revising these and adding more. It is not clear why elderly is a key word, when patients > 15 were included in this study
Introduction
· A thorough description of the error classification would be helpful here (could move this up from the methods).
· The final paragraph needs to be reworded and RPC better explained since this is integral to the study – perhaps this could be achieved by moving some of the detail in the ‘study setting’ section up.
Methods
· A brief description of admin/route/dose/patient errors would be useful instead of just referring the reader to a previous study
· The inclusion criteria has therapeutic errors caused by health care providers in a hospital setting, but then the next sentence defines therapeutic error differently, including that administered by medical personnel or by lay person. So it’s not clear how errors were defined for this study.
· In the methods, anything not related to what was done in the present study should be moved elsewhere in the manuscript.
· Statistical analysis – why did the authors choose to do a series of standalone pairwise comparisons instead of a multifactorial ANOVA? What was the level of significance? Were adjustments made to the alpha level in order to reduce the chance of a Type 1 error?
Results
· take care with wording throughout e.g. After saying that five patients died, there is no need to state that the remaining patients survived
· ‘One dead patient was treated with the wrong drug…’ sounds like a deceased patient was given a treatment
Discussion
· ME reporting is voluntary, which is mentioned as a limitation but should be discussed in more detail. Does this mean the more serious MEs are more likely to be reported? Do the authors have any estimate for how many MEs are not reported? If they happen in hospital, wouldn’t there be mandatory reporting somewhere?
· The finding that most errors happened during the afternoon shift – are the authors confident that there are indeed more errors at this time of day, or could it be related to staff having more time to report on the RPC compared with other times of day/night?
· Lookalike vials was discussed as a potential contributor for these MEs. In the drugs involved in MEs in this study, were the similarity of vials objectively compared?
Conclusion
· It is very well known that MEs can cause severe outcomes and even death in some patients – the conclusion needs to be adjusted to make clear what knowledge this study adds.
Tables
· Please be specific about what the values being reported are (N, %, mean (SD), and so on)
· Table 6 – ‘hospital stay after ME’ - is this because the patients died? If so, this should be reworded
I have made some suggestions for sections that need to be reworded above, but I suggest that the authors review the manuscript to ensure appropriate use of English language.
Round 2
Reviewer 2 Report
Dear Authors,
you have not included in your manuscript the figures illustrating your results that I requested. However, the manuscript after proofreading seems to be slightly better and addresses an important aspect of medical errors. Therefore, I will suggest to the editors to accept the article.
Kind regards.
Author Response
The responses to the reviewer comments
Editor and Reviewer 2
On behalf of my coauthors, I would like to thank you and all reviewers for the thoughtful review given to our manuscript. We very much appreciated your and reviewers’ critiques and comments on our manuscript. We have now revised according to your and reviewers’ suggestions and comments for our manuscript in a point-by-point fashion and discuss what changes, if any, have been made in the manuscript as a result of the review. For your convenience, our responses follow each of the reviewers’ comments and the revised words or sentences will be in the track changes form and the clear form. We highlighted the sentences that were revised according to reviewer’s comment with yellow color.
We look forward to hearing from you regarding our submission. We would be glad to respond to any further questions and comments that you may have.
Best regards,
Satariya Trakulsrichai
Reviewer 2
Dear Authors,
you have not included in your manuscript the figures illustrating your results that I requested. However, the manuscript after proofreading seems to be slightly better and addresses an important aspect of medical errors. Therefore, I will suggest to the editors to accept the article.
Kind regards.
Reply: Thank you very much for your suggestion. We have revised the manuscript by adding the figure 1 and 2 in the manuscript.